# Global lake thermal regions shift under climate change

Stephen C. Maberly [1✉], Ruth A. O'Donnell [2], R. Iestyn Woolway [3], Mark E.J. Cutler [4], Mengyi Gong [2,5], Ian D. Jones [1,6], Christopher J. Merchant [7,8], Claire A. Miller [2], Eirini Politi [4], E. Marian Scott [2], Stephen J. Thackeray [1] & Andrew N. Tyler [6]

Water temperature is critical for the ecology of lakes. However, the ability to predict its spatial and seasonal variation is constrained by the lack of a thermal classification system. Here we define lake thermal regions using objective analysis of seasonal surface temperature dynamics from satellite observations. Nine lake thermal regions are identified that mapped robustly and largely contiguously globally, even for small lakes. The regions differed from other global patterns, and so provide unique information. Using a lake model forced by 21st century climate projections, we found that 12%, 27% and 66% of lakes will change to a lower latitude thermal region by 2080–2099 for low, medium and high greenhouse gas concentration trajectories (Representative Concentration Pathways 2.6, 6.0 and 8.5) respectively. Under the worst-case scenario, a 79% reduction in the number of lakes in the northernmost thermal region is projected. This thermal region framework can facilitate the global scaling of lake-research.

[1] UK Centre for Ecology & Hydrology, Lancaster Environment Centre, Lancaster LA1 4AP, UK. [2] School of Mathematics and Statistics, University of Glasgow, Glasgow G12 8QQ, UK. [3] Centre for Freshwater and Environmental Studies, Dundalk Institute of Technology, Dundalk A91 K584, Ireland. [4] School of Social Sciences, University of Dundee, Dundee DD1 4HN, UK. [5] British Geological Survey, Keyworth, Nottingham NG12 5GG, UK. [6] Biological and Environmental Science, University of Stirling, Stirling FK9 4LA, UK. [7] Department of Meteorology, University of Reading, Reading RG6 6AL, UK. [8] National Centre for Earth Observation, University of Reading, Reading RG6 6AL, UK. ✉email: scm@ceh.ac.uk

Temperature is a global environmental variable of great importance as it controls a wide range of biological and geochemical states, structures and processes. In fresh waters, these include effects on organism size[1], species distribution[2], growth rate[3], opening of thermal niches for non-native species[4], food web interactions[5], phenology[6], stoichiometry[7], greenhouse gas fluxes[8] and metabolic rate and balance[9]. Lake classifications based on characteristics, such as mixing[10] or trophic state[11], are widely used and valued, but a global classification of lake thermal regions, although long-desired[12], is lacking, unlike in terrestrial and marine systems[13,14]. This lack of a classification framework restricts our ability to scale and contextualise lake research on a wide range of issues. A current pressing example concerns the effects of climate change. Among the most pervasive and concerning consequences of climate change on lakes are the direct and indirect effects of lake temperature including warming, reduction in ice cover, altered stratification and decreased internal nutrient loading[15–18] with consequences for all the states, structures and processes mentioned above.

In this study we produce a global lake thermal region framework, based on seasonal surface water temperature records from 732 lakes over sixteen years derived from satellite data. We use a lake model to expand the framework to all areas globally where lakes are present at a 2° grid and an App that produces a thermal region classification using locational or in situ data. We used a lake model forced by climate change scenarios to project how these thermal regions may shift in the 21st century.

## Results

**Lake thermal regions**. To produce a thermal classification scheme, we analysed satellite-derived lake surface water temperature data, twice a month, from 1996 to 2011 from 732 large lakes (available from the ARC-Lake database, www.laketemp.net; see Methods). Seasonal patterns of lake surface water temperature were analysed using a b-spline modelling approach that took ice-cover into account. K-means clustering produced nine coherent groups of lakes (Fig. 1a). Lakes within each group had a distinct and coherent pattern of annual mean temperature and annual mean range (Fig. 1b, c, Supplementary Figs. 1 and 2) and clear latitudinal distributions (Fig. 1d). This allowed them to be assigned descriptive binomials based on their seasonal temperature cycle (Fig. 1 and Table 1). Five groups were found exclusively in the northern hemisphere (Fig. 1e). The Northern Temperate lakes showed the largest mean seasonal lake surface water temperature range of 23.6 °C with one lake having an average range of over 29 °C. Between 90 and 100% of Northern Frigid, Northern Cool and Northern Temperate lakes, were ice-covered with a duration of 61, 50 and 29% of the year, respectively (Table 1). Tropical Hot lakes had the highest mean temperature (29.0 °C) and the lowest annual range (3.9 °C) and 76% occurred within 15° of the equator. Lakes assigned to the Southern Warm and Southern Temperate thermal regions were found only in the southern hemisphere, and were distinguished from the northern hemisphere lakes by their lower temperature range for a given mean temperature, rather than by the phase of their seasonality. This low temperature range results from the effect on air temperature of the proportionally lower land area compared to the ocean area in the Southern compared to the Northern Hemisphere[19]. Lake elevation had an effect on the distribution of the thermal regions: for example, the two most northerly lake thermal regions, Northern Frigid and Northern Cool, were also found at high elevation in the Tibetan Plateau at a relatively low latitude. Accordingly, apart from Northern Warm, a given thermal region occurred at higher elevation near the equator than at high latitudes (Supplementary Fig. 3).

**Predicting seasonal patterns across the globe**. In order to expand the global distribution of lake thermal regions from the 732 lakes to the entire land area where lakes are present, the physical lake model FLake[20] was applied globally at a latitude-longitude resolution of 2° ($n = 4221$ grids) using the average lake depth for each cell based on the HydroLAKES database[21]. The model was driven by forcing data from a global reanalysis, ERA-Interim (see Methods) over the period 1996–2011 that matched the ARC-Lake period and included surface air temperature, wind speed, solar radiation, thermal radiation and specific humidity. The overall global-mean lake-surface temperature was 14.1 °C and the modelled annual mean lake surface temperatures followed the expected latitudinal pattern (Fig. 2a). The mean annual range of temperature was 16.1 °C (Fig. 2b) and at many locations the range was orders of magnitude greater than long-term temperature change[16] or diel changes[22]. The modelled data were analysed using the rules derived from the analysis of the ARC-Lake data (see Methods), to produce a global map of lake thermal regions (Fig. 2c), with posterior probability of thermal region membership shown in Supplementary Fig. 4. The largest number of lakes in the HydroLAKES database was in the Northern Cool thermal region (40.4%), followed by Northern Frigid (37.8%) and Northern Temperate (11.2%); all the other groups comprised less than 5% of the global total. The Northern Cool thermal region covered the greatest area (25.2%), followed by Northern Frigid (21.0%) and Tropical Hot (13.4%) (Table 1). The thermal region of each of the 1.4 million lakes in the HydroLAKES database is available as a downloadable file (see "Data availability"). To facilitate and promote future use of these lake thermal regions, an R Shiny app has been produced that allows a specific lake to be categorised to one of the nine lake groups, with an estimate of uncertainty, by entering either location by clicking on a map or by entering the latitude and longitude or by uploading in situ water surface temperature measurements. The code for this app is available at GitHub (see "Code availability"). The RGB code for each thermal region is given in Supplementary Table 1.

**Relating lake thermal regions to other global schemes**. The broad relationship between lake surface water temperature and air temperature[23] (Supplementary Fig. 5) suggests that the lake groups may correspond to air temperature, existing climate zones or terrestrial ecoregions, and thus be redundant. However, neither the 11 major Koppen–Geiger (K–G) climate classifications based on the main climate groups and temperature[13], nor 13 of the 14 (excluding mangroves and tropical and subtropical coniferous forests and combining them as Other) Terrestrial Ecoregions of the World[24] were uniquely linked to the lake thermal regions (Fig. 3a, b). Similarly, lake thermal regions did not match uniquely to the nine air temperature groups produced using available datasets and the same statistical technique (Fig. 3c). This indicates that global patterns of lake surface water temperature dynamics are not effectively captured by air temperature or existing terrestrial ecology classification schemes, and that the drivers of lake surface water temperature differ from those of climate and vegetation type. The low spatial agreement between surface lake water temperature and air temperature, matches the observed lack of temporal rate of change in air and lake temperature at individual locations[16].

**Applicability of thermal regions to small lakes**. The lakes in the primary dataset were typically large because of the spatial resolution of the satellite data: the smallest lake was 49 km² and the median surface area was 242 km². However, 99% of lakes globally (down to a cut-off of 0.1 km²) are less than 10 km² in area[21]. Lake area is known to influence various components of

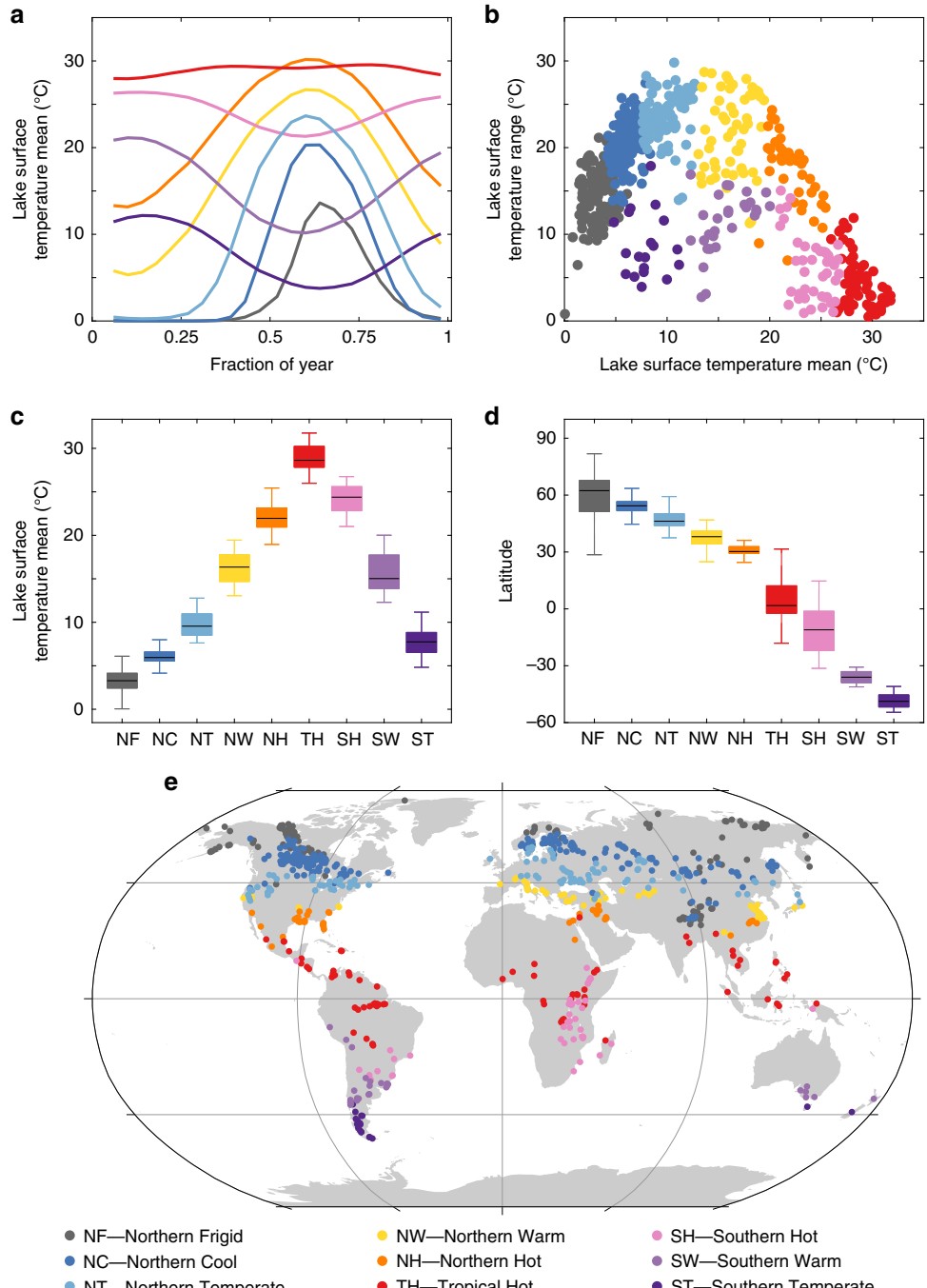

**Fig. 1 Allocation of studied lakes to thermal regions. a** Mean seasonal satellite-derived lake surface water temperature. **b** Annual range vs. annual mean for the individual lakes. **c** Boxplots of temperature in each lake group, the box is the interquartile range, the horizontal line is the median and the whiskers are 1.5 times the interquartile range. **d** Boxplots of latitude in each lake group. **e** Map of studied lakes by thermal regions.

lake heat budgets including effects of atmospheric stability[25], wind speed[26] and turbulent surface fluxes[26] as well as the diel cycle of surface temperature[22] suggesting that global seasonal temperature dynamics of large lakes might not necessarily be representative of small lakes. The applicability of the classification to smaller lakes was assessed by compiling in situ data on seasonal surface temperature from 71 lakes with a median surface area of 0.9 km$^2$ (Supplementary Table 3). These were allocated to lake thermal regions using the in situ data and also based on their location in the ERA-Interim global-gridded map (Fig. 2c). Totally, 80% of the lakes were correctly matched to the same thermal region based on geographic location and this

increased to 97% if the second closest match was included (Fig. 3d). This indicates that the thermal region scheme is applicable to small as well as large lakes. The 732 lakes with satellite-derived surface temperature data include lakes that are brackish or saline as well as fresh. Except that overlying meteorology will influence both the likelihood of salinity and the surface temperature cycle, we do not expect impacts of salinity to supersede meteorological effects sufficiently to distort greatly the classification. In mainland Australia, for example, clustering analysis showed that all the lakes in the dataset fitted the Southern Warm class (Fig. 1e) despite having a large range of salinities.

**Table 1 Details of the nine thermal regions. The number of lakes refers to the HydroLAKES database and the historic period.**

| Thermal region name | Name code | Number of lakes (% of total) | Percent coverage of thermal region | Mean temperature (mean range, °C) | Percent of time ice-covered (% of lakes with ice cover) |
|---|---|---|---|---|---|
| Northern Frigid | NF | 575,983 (37.8) | 21.0 | 3.3 (14.4) | 61 (100) |
| Northern Cool | NC | 539,638 (40.4) | 25.2 | 6.1 (20.7) | 50 (99.6) |
| Northern Temperate | NT | 159,661 (11.2) | 13.0 | 9.8 (23.6) | 29 (91.9) |
| Northern Warm | NW | 24,185 (1.7) | 7.0 | 16.3 (21.8) | 1.0 (9.0) |
| Northern Hot | NH | 25,369 (1.8) | 5.9 | 22.1 (17.4) | 0.0 (0.0) |
| Tropical Hot | TH | 46,245 (3.2) | 13.4 | 29.0 (3.9) | 0.0 (0.0) |
| Southern Hot | SH | 25,131 (1.8) | 8.3 | 24.2 (6.1) | 0.0 (0.0) |
| Southern Warm | SW | 24,191 (1.7) | 5.9 | 15.7 (11.4) | 0.0 (0.0) |
| Southern Temperate | ST | 6745 (0.5) | 1.3 | 7.9 (8.7) | 5.0 (21.1) |

**Lake thermal region response to projected climate change.** Analysis of historic data has shown that summer lake surface temperature is increasing globally but varies spatially depending on lake characteristics and regional changes in climate[16]. Similarly, forecast loss of ice cover by the end of the 21st century in lakes of the Northern Hemisphere is spatially variable[18]. To project future changes in the identified global lake thermal regions, the globally gridded lake model was forced by four bias-corrected climate projections under three representative concentration pathway (RCP) scenarios (RCP 2.6, 6.0 and 8.5), available from the inter-sectoral impact model intercomparison project (ISIMIP; see Supplementary Information). These global projections of lake thermal regions were also compared to those simulated by FLake forced with ERA-Interim (1985–2005) to ensure that no bias was introduced by using different global climate models. No differences were observed between the simulated lake thermal regions by forcing FLake with using climate data from ERA-Interim or those from the ISIMIP projections.

To illustrate projected future changes in the lake thermal regions, we present results from the lake model forced with HadGEM2-ES, but the other climate projections produced very similar results (Supplementary Figs. 6–9). There will be major changes in the global distribution of lake thermal regions by 2080–2099 (Fig. 4). Higher latitude lakes will tend to become like lower latitude lakes. The frequency of these shifts is projected to increase with the severity of climate change: 12, 27 and 66% of all lakes will move to a lower latitude thermal region for RCP 2.6, 6.0 and 8.5, respectively. For RCP 8.5, the number of Northern Frigid lakes will decline by 79%, while the number of Northern Temperate and Northern Warm lakes will increase by 166 and 228%, respectively. The status of a total of 13,715 lakes (1%) will shift by more than one thermal region and an additional 48 Northern Cool lakes will shift status by three thermal regions to become Northern Hot lakes.

## Discussion

We show here that robust, statistically derived lake thermal regions can be produced. This framework will strongly complement and underpin analyses of the global stock of lakes, and help in the upscaling of a range of characteristics that are affected by temperature. It will also allow effects of climate change on a range of important physical processes and states to be contextualised including ice-cover[18] and altered mixing regimes[27]. Synoptic data derived from remote sensing are powerful approaches to characterise large scale patterns in surface conditions of lakes. However, a full assessment of climate change effects on lake structure, function and conservation value will require additional data derived from traditional and high-frequency measurements of conditions over the full depth of a lake. This will be especially the case for specific lakes or lake regions where an additional range of attributes including the connectivity between the lake and its catchment, lake bathymetry and biodiversity will be required,

especially for key species that are ecological engineers or of high conservation or commercial value.

The important ecological goods and services that lakes provide may be compromised if increasing temperature promotes potentially harmful algal blooms[28,29] and modifies species composition[2]. For example, there is increasing concern about the loss of cold-water fish species many of which are not only ecological engineers but also of economic importance, such as the salmonids[30]. Our results on climate change effects suggest that northern high latitude lake thermal regions, where salmonids thrive, will be particularly susceptible to future global warming[31]. This supports the recent demonstration using historical long-term records that there is already a widespread loss of ice from high latitude lakes in the northern hemisphere and that this is projected to increase in the future[18]. The analysis of O'Reilly et al.[16] showed that trends in summer surface warming were not regionally clustered. Instead, a range of climatic, geographic and morphometric data were able to distinguish between eight different groups using a regression tree approach[16]. In agreement with this we found a low similarity between these clusters of summer temperature trend (defined by the O'Reilly et al.[16], Table S4) and the thermal regions defined here (Supplementary Fig. 10).

There is growing interest in defining systems in terms of biomes to help describe large scale patterns and predict how they may change[32]. Surface water temperature will not be the only determinant of a lake biome, however, it is an important component that with additional globally available data produced by Earth Observation and large scale compilations of freshwater species biodiversity[33] will provide the opportunity to move toward characterising lake biomes.

## Methods

**Fitting the data to smooth functions and grouping the lakes.** We analysed 16 years (1996–2011) of satellite-derived lake surface water temperature data, available twice a month from 732 lakes across the globe, derived from the ARC-Lake dataset produced by the (advanced) along-track scanning radiometers[34] at (www.laketemp.net)[19]. These represented the lakes with complete temperature records and that matched the lakes within the GloboLakes database[35]. Lake-wide average surface temperatures were calculated[36].

Smoothing functions were fitted to each lake-mean time series to capture the major temporal patterns in the data using a saturated b-spline based statistical modelling approach that places a knot at every time point[37]. When ice was detected, either by reflectance imagery or because of sub-freezing surface temperatures, the water below the ice was set to 0 °C. To accommodate the discontinuities in the smooth time series that result from ice cover, the coefficients of the basis functions centred on the knots that correspond to times with ice cover were set to zero. This effectively drops out the basis function whenever ice is present (Supplementary Fig. 11). This approach ensures that the number of basis functions used to estimate the underlying curve is the same for each time series which is necessary for the functional clustering of the curves. A data-driven approach was used to aggregate the individual lake records into coherent groups using K-means clustering. This was applied to the b-spline basis coefficients to identify groups of lakes with a similar temporal pattern (underpinned by the shape of the seasonal pattern, the mean temperature and any long-term change over time, if present). Nine groups were identified as being statistically optimal using the gap

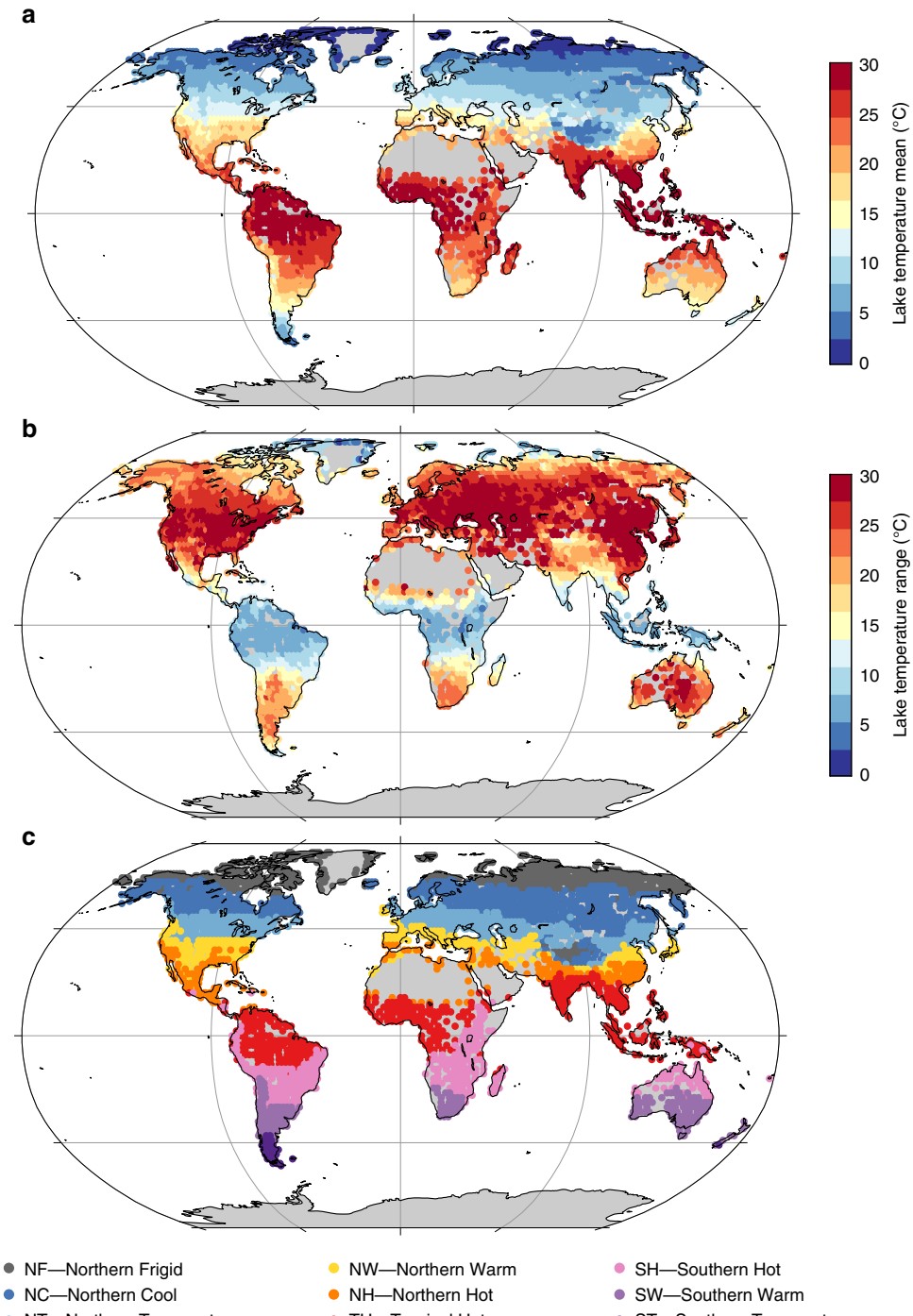

**Fig. 2 Modelled lake surface water temperature dynamics. a** Mean lake surface water temperature. **b** Seasonal range in lake surface water temperature. **c** Map of the nine lake thermal regions. Locations represent 2° grids where lakes exist based on the HydroLAKES database (see methods); grey areas are where lakes are absent in the database. Lake surface temperatures and the thermal regions were calculated by the lake model driven by ERA-Interim data from 1996 to 2011.

statistic[38] although numbers of groups between eight and eleven also produced reasonable fits to the data with lake membership of the bulk of the groups remaining robust to group number.

**Predicting seasonal patterns across the globe.** In order to expand the geographical applicability of the data, a classification rule was developed based on the original groups. Functional principal components analysis (FPCA)[39] was applied to the 732 smooth seasonal curves to reduce the dimensionality of the data while capturing the vast majority of the underlying variability. Two FPCs captured 97.3%

of the variability in the curves enabling the smooth average seasonal patterns (seasonal lake temperature profiles) to be represented in two-dimensional space (Supplementary Fig. 12). The use of FPC scores for classification, rather than other univariate temperature characteristics such as seasonal amplitude or average, and lake characteristics such as elevation, latitude and longitude, allows much more subtle and complex variations in the seasonal patterns of the groups to be taken into account. Quadratic discriminant analysis (QDA) was used to establish a classification rule for these lake temperature patterns using their associated nine group identifiers. For a lake, with at least 1 year of data, a smooth curve for the 'seasonal' time series can be projected onto 2D space using the FPC projection to

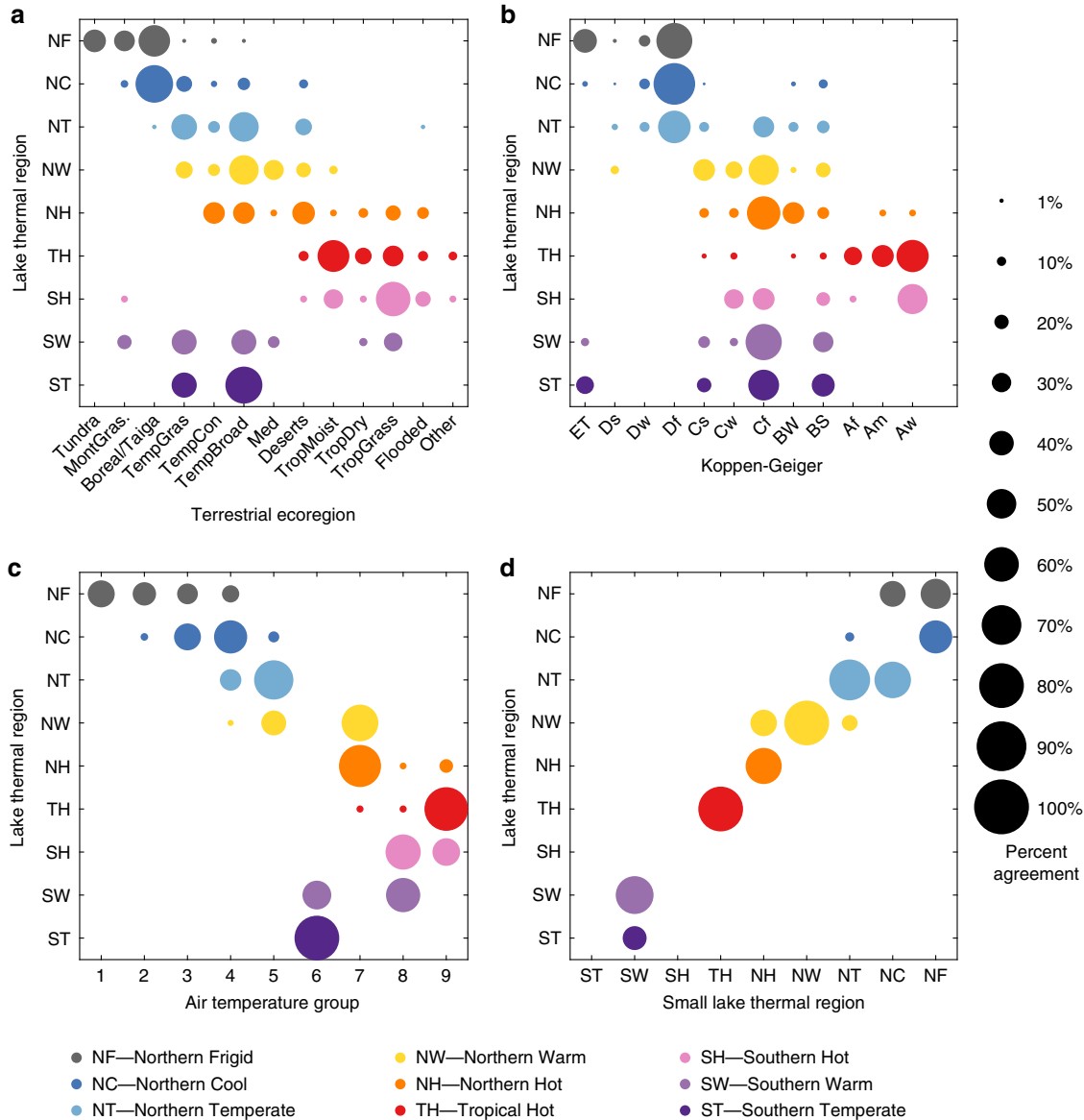

**Fig. 3 Relationship between thermal regions and other global patterns. a** Terrestrial ecoregions of the world (see ref. [24] Supplementary Table 2). **b** Koppen–Geiger binomial categories (see ref. [13] Supplementary Table 2). **c** Air temperature categories (see methods). **d** Data from small lakes (x-axis) compared to geographical location (y-axis) based on the global thermal regions identified from the lake model forced with ERA-Interim (Supplementary Table 3). The colours represent the lake thermal regions (as shown in Fig. 1) and the size of the circle the percent of lakes from a thermal region across the categories of the other characteristic.

produce scores, and this was then used to classify the lake into one of the nine groups along with an estimate of uncertainty. Using this approach, 96% of seasonal curves from the original 732 lakes were assigned to their correct group.

**Lake temperature model**. To generate a global map of surface temperature groupings, we simulated lake surface water temperatures worldwide with the lake model, FLake[20]. FLake has been tested extensively in previous studies, including detailed validations across a spectrum of lake contexts[27]. Here, we used Flake 'out of the box' without calibrating the model parameters to improve fit. The meteorological variables required to drive FLake are surface air temperature, wind speed, surface solar and thermal radiation, and specific humidity. A set of external parameters are also required to drive FLake. These include mean depth, which in this study was extracted from the HydroLAKES database[21], the light attenuation coefficient, which was set to 1 m$^{-1}$ and fetch that was estimated as the square root of surface area, derived from the HydroLAKES database. For the ARC-Lake period (1996–2011) we ran the lake model using atmospheric forcing from the ERAav-Interim reanalysis product[40]. We selected data from the grid point situated closest to the centre coordinate of each lake. For the 732 lakes with satellite-derived temperatures, FLake simulated accurately various features of the surface

temperature annual cycle when compared to observed satellite-derived lake surface temperatures. The average (across all lakes) mean absolute difference (MAD) from the ARC-Lake observations was 0.76 °C. Over 80% of the lakes were simulated with a MAD of <1 °C. In order to simulate seasonal temperature dynamics of lakes across the Earth's land surface we used FLake, driven by the ERA-Interim data, for each 2° latitude-longitude grid, where the mean depth for each grid was calculated as the average depth of all lakes within a given cell, as calculated from the HydroLAKES database[21], and the light attenuation was set as before. This set-up is common in global-scale simulations of lake surface water temperature and is often used, for example, for lake representation in numerical weather prediction[41,42]. Although light attenuation can affect the mixing regime[43], the effects on seasonally averaged surface temperature is relatively small[44] compared to the differences between thermal regions. The greater impacts on surface temperature, especially in clear, relatively small lakes suggests such lakes could show some deviation in thermal region from neighbouring lakes.

To evaluate lake thermal region responses to future climate change, FLake was driven by air temperature at 2 m, wind speed at 10 m, surface solar and thermal radiation, and specific humidity from bias-corrected climate model projections from the ISIMIP2b (https://www.isimip.org/protocol/#isimip2b)[45]. Projections from GFDL-ESM2M, HadGEM2-ES, IPSL-CM5A-LR and MIROC5 for historic

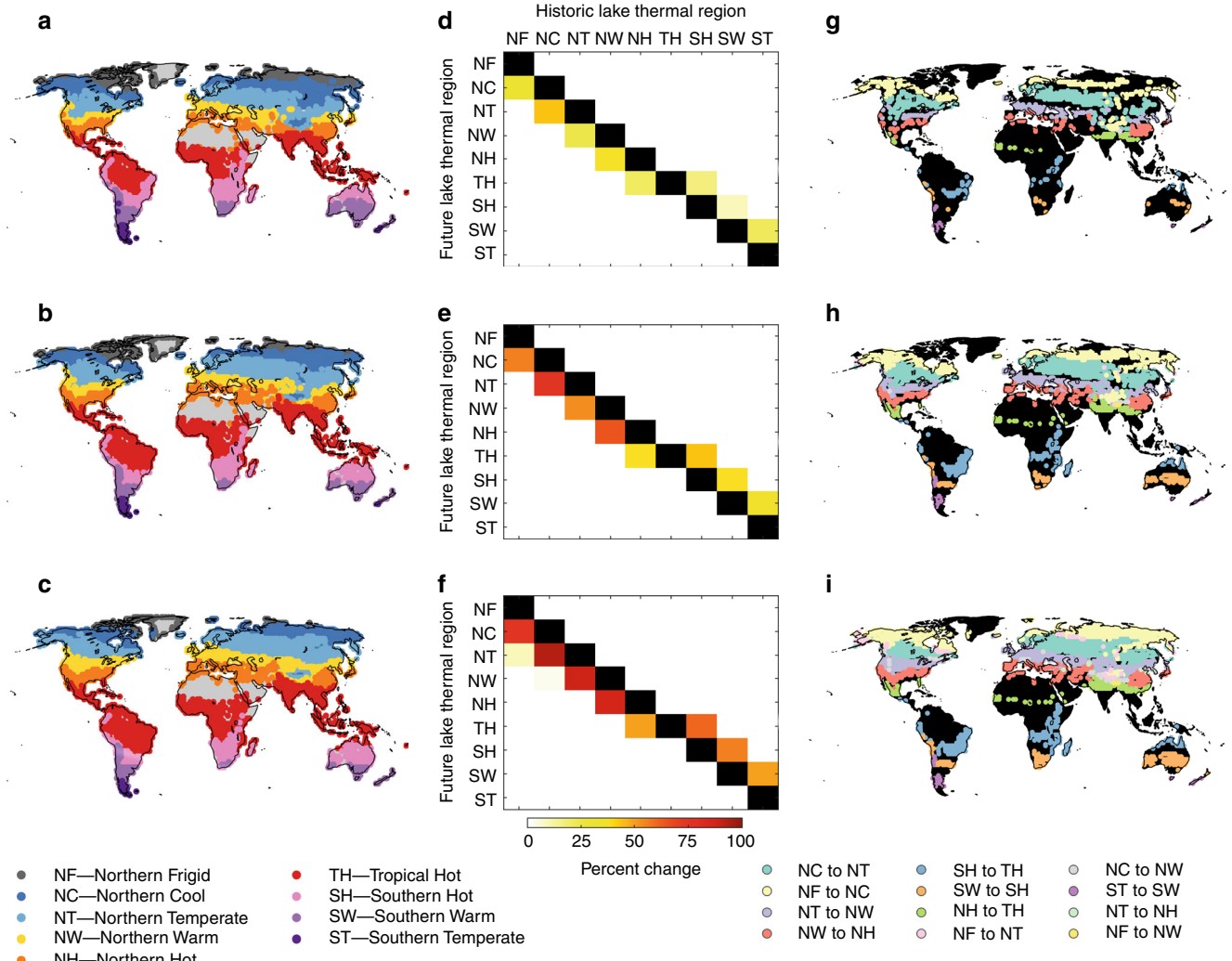

**Fig. 4 Change in thermal region distribution for three climate scenarios. a, d, g** RCP2.6; **b, e, h** RCP6.0; **c, f, i** RCP8.5. **a–c** show forecast distribution of lake thermal regions (grey represents areas without lakes); **d–f** show percent of lakes in each thermal region that will change by 2080–2099 compared to the historic period (1985–2005); **g–i** show lakes that will change thermal region compared to the historic period (black represents areas where no lakes have changed), the different changes in the legend are ordered by decreasing frequency of occurrence. All results, including those from the historic period, are produced using the bias-corrected HadGEM2-ES projections.

(1985–2005) and future periods (2080–2099) under three scenarios: RCPs 2.6, 6.0 and 8.5 were used. The lake model outputs were classified using the FPCA-QDA approach, above, to produce a global map of lake groups along with their uncertainty.

**Relating the lake classification to other global schemes**. We categorised the lakes by other global schemes in order to test whether or not the lake thermal regions defined here offer unique information. For the K–G climate classification[13], the first five main climate groups subdivided further in some cases by the third classification based on temperature, producing eleven categories in total. For the terrestrial ecoregions of the world[24], 13 ecoregions were used since Mangroves and Tropical and subtropical coniferous forests were only relevant to 4 lakes and these were combined as Other. Finally, the lake thermal regions were compared to temperature clusters based on bi-monthly air temperature data for the 732 lakes between 1995 and 2012 derived from the Climate Research Unit Time Series v.3.22[46]. The air temperature clusters were produced in the same way as described for lake surface water temperature except that there was no need to impose discontinuities caused by ice cover. For air temperature, a grouping of six clusters was statistically optimal but nine clusters, matching the number of lake clusters, was similar statistically and so we chose to use this number of clusters. These nine air clusters were identified with numbers, ordered to match the lake temperature thermal regions as closely as possible

**Description of the App and its use**. An app has been developed using R Shiny[47], which displays the lake thermal eco-regions across the globe. Within the app, a user can obtain the predicted thermal region for a specific lake location along with the corresponding predicted seasonal temperature pattern and confidence in classification. The confidence in classification is determined by the posterior probability of thermal region membership. The lake location can be specified either by clicking on the map or by entering a set of coordinates. In addition, the app enables the user to upload their own time series of in situ lake temperature data to produce a predicted thermal region classification and confidence in the classification; the app provides full details of the data format required. For longevity, the code behind the app is stored on GitHub (see "Code availability").

## Data availability
Satellite lake temperature data are available at http://www.laketemp.net.

## Code availability
The lake model used is available to download at http://www.flake.igb-berlin.de/. The code to predict the thermal region of a lake based on location or in situ data is available as an R Shiny app at GitHub: https://github.com/ruth-odonnell/LakeThermalRegions/. We will also maintain the app for as long as possible at https://shiny-apps.ceh.ac.uk/LakeThermalRegions.

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

## Acknowledgements

We thank Boris Adamovich, John Anderson, Monica Diaz, Arni Einarsson, David Hamilton, Mark Hoyer, Thora Hrafnsdottir, Helen Kettle, Biel Obrador, Andrew Paterson, Michael Paterson, Simon Patrick, Baoli Wang and Hai-Jun Wang for providing unpublished in situ lake surface temperature data. The work was supported by Globo-Lakes funded by the Natural Environment Research Council (grant number NE/J021717/1). RIW received funding from the European Union's Horizon 2020 research and innovation programme under the Marie Skłodowska-Curie grant agreement No. 79181 and the H2020 project EUSTACE (grant number 640171). Generation of the ARC-lake dataset was funded by the European Space Agency (contract 22184/09/I-OL).

## Author contributions

S.C.M., R.D. and R.I.W. conceived the idea for the paper. R.D., C.A.M. and E.M.S. led the statistical analyses. M.E.J.C. and E.P. compiled the catchment data. R.I.W performed the modelling. S.C.M and R.I.W led the drafting of the paper. All were involved in early discussions on the focus of the work and provided critical feedback on drafts of the paper and approved the final version (S.C.M., R.D., R.I.W., M.E.J.C., M.G., I.D.J., C.J.M., C.A.M., E.P., E.M.S., S.J.T. and A.N.T.).

## Competing interests

The authors declare no competing interests.
