## [Peer Review File · Nature Communications]

Reviewers' comments:

Reviewer #1 (Remarks to the Author):

There is currently no global lake classification scheme for temperature, perhaps the most important variable in ecosystem function for lakes. This knowledge gap is probably because such an exercise is more complex than simply relating air temperature to water temperature. Here the authors have included not only average water surface temperature but also the seasonal dynamic, which is critical for understanding ecosystem function. Temperature and the seasonal patterns of turnover and stratification have broad and profound effects on biogeochemistry, gas flux, trophic relationships, organismal metabolism, and everything else that readily comes to mind in limnology. So the subject matter here is important, and the result is exciting. Further it's very useful to see that the authors have 1) used the classification scheme to consider climate scenarios, and also 2) demonstrated that small lakes can be correctly placed in ecoregions using extrapolation from the satellite data. The restriction of satellite data to large lakes has been a major criticism of its use.

Thus the authors have filled a knowledge gap that will be of wide interest for the aquatic science/management community. I would hope that it would also be noted by other environmental scientists studying climate change who need to incorporate water bodies in their analyses, e.g. for gas flux and atmospheric dynamics.

I have 3 major suggestions, and various suggestions by line number.

1) Given that the research community at large could indeed use this work to feed global or regional analyses well outside of limnology – e.g. for atmospheric dynamics, or gas flux – researchers would need to trust wholly that the analyses are robust, such that they could view it as a product ready to plug and play, without having detailed limnological expertise themselves. In the Supplementary Methods (Line 28) it looks like the novel methods for accounting for periods of ice cover were not included for the reviewers, and this is a concern. If Gong et al. (in prep) methods are rejected by reviewers, or subsequently revised, what are the implications for this lake classification scheme? Ice covered lakes are indeed those anticipated to experience greatest change (Northern Frigid lakes on main text line 125-126). I have no way of evaluating the seriousness of this concern with the material at hand.

2) The authors could usefully place their findings in the context of those of O'Reilly et al. 2015 which produced groupings based on temperature trends, or Sharma et al's 2019 evaluations of ice cover (fig. 1) and projections of ice loss (fig. 2). Currently those papers are referenced but no explicit comparisons of results are made that would synthesize this paper's findings with those of previous work. Were the groupings or other spatial patterns similar? Did groupings of this paper map onto long terms trends shown in O'Reilly et al (e.g. their figure 3)? In particular, O'Reilly et al's finding that warming was not predictable by region, lake depth, or air temperature trends, but related

additionally to cloud cover and radiation trends, together might suggest that the projections of ecoregion change in this paper will be in error. In a practical sense, some of the highest and lowest warming rates in that paper occurred in lakes that were of similar elevations and air temperatures but with divergent cloud cover or solar radiation temporal trends. Data from both papers are all public so there really is not much constraint on how far the authors could take a synthesis.

3) Several of my comments are meant to improve the reader's understanding of the motivation for the paper. Right now the text does not establish the motivation very strongly - this weakness undermines the important messages in this paper. The summary is the primary section needing revision in this respect.

Line 23- the reader doesn't know what a lake thermal ecoregion is – limnologists don't currently have them, and may not have thought much about needing them

Line 24 – the reader doesn't know what an ecoregion framework is yet, so using it as a baseline is pretty confusing

Line 32 – RCPs – this is not familiar terminology to many aquatic scientists – something like “recent scenarios of emissions” might get to the point more quickly

Line 97-99 – the result that water temperature doesn't track air temperature neatly is very similar to a main finding of O'Reilly et al. 2015 (Figure 2), one of several places in this manuscript where this work could be more soundly placed in the context of previous global analyses.

Line 115 – to say that O'Reilly et al. found regional variations is somewhat misleading – the trends didn't typically clump by region but by a variety of regional climate and geomorphic factors – many of the groupings were dispersed throughout the world rather than by region (e.g. Fig 3 <https://agupubs.onlinelibrary.wiley.com/doi/full/10.1002/2015GL066235>).

Line 122-125 – this sentence doesn't quite make sense – the lakes don't move. Something more like... “The ecoregion status of a lake will shift toward those that are currently represented at lower latitudes...”? It's important to get this statement right since readers need to think about high latitude lakes becoming more like low latitude lakes over time, in order to grasp the concept and consider the implications.

Line 127 – use of the “lakes... move” again – I'd suggest rethinking this language. The reader is in a spatial state of mind for this paper, so “move” conjures up an image of the lake moving on a map.

Line 130-32 – I have to think about it pretty hard to understand this statement that “seasonal surface temperature dynamics among lakes are essentially a continuum”. Is this a really important concept to convey? If so, I'd suggest rephrasing.

Line 136-138 – this is an important starting point but for many of these issues we will need to know more about lake morphometry, connectivity with other surface and ground waters, biodiversity, and probably other components that will affect refugia and biogeochemistry. A big unknown when working with satellite data is the focus on surface temperature and an inability to infer important characteristics of stratification and deep water dynamics, let alone biodiversity that is not observable at the surface. While I don't think the authors need to caveat it to death, some acknowledgements of these important limitations should be included. Personally I am always concerned that the erroneous impression that satellite-based modeling can capture all of the important variables now when they actually capture relatively few aspects will lead to undesirable outcomes – e.g., that in situ monitoring programs will be terminated by decision makers, or well-meaning graduate students will restrict their analyses to satellite data and associated modeling rather than understanding limnological data in 3 dimensions. (Maybe someday, but right now we are not there.)

Figure 2 – it looks like grey not black indicates missing data

Figure 3 – without a legend for all the detail on these axes, panels b and c aren't useful, and panel a is marginally more interpretable. A 1:1 line would be useful in panel D where one could think a little more about ecoregions where small lakes might not map as well.

Figure 4 – in the caption, it would be useful to note that black on the middle column is 1:1. What does black mean on the far right column? Presumably grey on the far left means missing data?

Supplementary Fig. 2 – was coherence quantified? If not, this figure does not literally show coherence, but it shows temporal patterns of lakes within each grouping (and the reader may visually infer some coherence).

Supplementary Fig. 3 and Line 62-66 – Lake elevation can't affect latitude of a lake – the caption needs to be revised somehow. I've been staring at the figure for some time now and the message remains unclear. I find the reference to the Figure in the main manuscript line 62-64 equally perplexing.

Altogether this manuscript represents a lot of work and I will be excited to see how the research community can build upon it!

Reviewer #2 (Remarks to the Author):

In general, lakes are considered to reflect their watershed, and have been loosely characterized based on their terrestrial biomes (e.g. alpine lakes), lake-specific characteristics (e.g. shallow lakes), or their water quality (e.g. eutrophic lakes). The paper proposes a quantitative approach that can characterize lakes into ecoregions based on their thermal properties (mean and variation). In itself, that is somewhat novel – ecoregions that are defined purely based on mathematics. In fact, it is quite astounding how distinct these patterns appear to be (Supplementary Fig 2). The broader community would find this interesting in a theoretical context at the very least.

The level of detail provided appears to be adequate for reproducibility and many of the analytical tools are accessible. I appreciated the efforts to make these results accessible and applicable via the online app. However, I could not get the online app to load, and this is obviously an important component of broader implications and outreach associated with this work.

Ultimately, I was comfortable with the analytical approach. Initially, I was concerned that the long term mean temperature was not a good approach given that other studies have shown that lakes have been warming over time, at least seasonally. However, the approach seems robust. Many of my questions were adequately addressed in the Supplementary Information. Although the ice-detection method is still in the process of peer review, I recognize that with 2 temperature measurements each month, the temporal resolution is at the scale where error in ice detection is unlikely to have a substantial impact on the long-term mean.

I was a bit concerned about accuracy of the projections. Most lakes are warmer than air (as can be seen in SI Fig 5) and other studies have found that many lakes are warming faster than air temperatures. Solar radiation is a major contributor to lake temperature, and global brightening and dimming is a factor contributing to lake warming trends. I wondered to what extent solar radiation has been factored in? If the projections are based solely on air temperature projections, this needs to be clear in the main manuscript and there should be some discussion about other contributing factors. I do not think that air temperatures are really that good of a predictor of lake temperatures.

Ultimately, while I was comfortable with the analytical approach that was used here, it is fundamentally a statistical approach to defining ecoregions. I was left wondering if it really held up with respect to the actual ecology of these systems. For example, in terrestrial biomes, temperate grasslands and deciduous forests have similar mean temperatures but different amounts of precipitation, and the ecological structures and communities are fundamentally different. In these lakes, are there real differences between the southern and northern warm lakes when the statistical differences are only with respect to seasonal temperature variation? On one hand, I was looking for more of this type of exploration into ecosystem structure and function; on the other hand, this paper provides a framework that would allow hypotheses about differences and similarities to be addressed in future work (and perhaps these authors are already considering this as a next step). At the same time, these lake ecoregions are not as useful as being proposed here if they don't clear

translate into meaningful ecological differences. At the very least, I am not sure if the term 'ecoregion' is appropriate when only one variable (temperature) is explored.

Supplementary Information

L62-63. Why was the light attenuation coefficient set to 1 m⁻¹? How much difference does it make if a different value is used? (e.g. Just because it is widely done for some types of modeling doesn't necessarily mean it's also the best approach here.)

L 76. What data were extracted from ISIMIP for the projections – all variables or just temperature?

Reviewer #3 (Remarks to the Author):

This article presents a comprehensive and novel analysis of the global distribution of lake surface temperature characteristics. Its broader aim is to provide a framework to help interpret knowledge of lake ecology and effects of global change. It further claims that future climate warming will shift a certain proportion of lakes from a higher latitude to lower latitude ecoregion, whereby the northernmost lakes will be particularly affected. I think the conclusions are well supported.

The study is based on a representative global set of satellite-derived lake surface/skin temperature data and is combined with global modelling of lake surface temperature using an established physical model forced by state of the art meteorological data and climate projections. The authors also provide an online interactive tool to classify lakes into "ecoregions" according to the analysis in the manuscript. The code and data documenting the analyses are available on github, which enhances future use of the data and the classification scheme within the scientific community. Altogether the online material presents a useful package to help others build on the authors' work.

I think this is a good article and in my opinion it represents a valuable and potentially high-impact scientific contribution. The "ecoregions" provide a useful framework for interpreting impacts of climate warming. The paper is concise, well written and well structured. In some places it is too concise and I have described these below. The combined modelling and observed data analysis is a robust approach. The authors have made an effort to be complete by including a separate analysis of lakes too small to be included in the satellite data analysis. I found the consideration of the effect of lake elevation on the thermal classification necessary and informative. The "ecoregions" are highly correlated with latitude but the authors provided an analysis of the added information of their scheme compared to other classifications or simply latitude. The statistical analyses appear to be thorough and appropriate, though some information is missing. The most important references have been reviewed to my knowledge. Almost all of my comments are minor and related to clarity or a wish for more details. Most of these things can be dealt with in the supplementary material.

There is only one potentially significant issue, which is possibly just a misunderstanding. In the global modelling, the authors state in the methods that they used ERA-Interim reanalyses to force the lake model, which is an established and acceptable method to run lake simulations. However I see a potential issue with the comparison of climate warming scenarios (RCP 2.6, 6.0 and 8.5) with historical data derived from ERA-Interim reanalyses to draw inferences about the effect of warming. This would introduce bias into the results, and any shifts of lakes from one ecoregion to another may be due in part to this bias rather than due to warming alone, which would question some of the results. The valid approach is to compare the RCP warming projections with the historical scenario generated from the same global climate model. It is possible that this is what the authors in fact did because supplementary figures 6, 7, 8, and 9 show a “historical” scenario. In any case this is not clear to me and I ask that the authors clarify whether the “historical” data are derived from ERA-Interim or from the respective climate model. If the historical scenario is based on ERA-interim, then additional work is required to deal with the bias.

Other comments

I am questioning the use of the term (thermal) “ecoregion”, which implies that lakes are classified according to their ecology whereas in fact they are classified only according to surface temperature. Surface temperature alone is insufficient to classify lake ecology. The authors gave a nice summary of how the thermal classification can contribute to classification of lake biomes (L147-151) but perhaps a different term is more appropriate. If you feel differently, it would be helpful to establish a broad link between lake ecology type and the “ecoregions” according to surface temperature based on literature, for instance. This is a semantic issue and I am not questioning the approach in general.

Would it be worthwhile to include a southern frigid region to include Antarctic lakes for the sake of completeness?

I am confused about how lakes were classified according to location (e.g. L 113, Supplementary Table 2). It is probably quite simple but not obvious to me from the manuscript. Please provide details about how the criteria were derived to classify lakes according to geographic location.

Could you comment on how you think the classification scheme works for brackish or saline lakes in arid or semi-arid regions? For instance many of the natural lakes shown in Australia are salty and/or even ephemeral. How did the lake model perform against satellite data in these regions?

Do you know why southern hemisphere lakes show less variability at the same mean temperature than northern hemisphere lakes?

L80: “was” not “were”.

L81: “all” repeated.

L113: This sentence requires some more explanation. What is meant by the sub-dominant group?

L123: “projected” is more appropriate than “forecast”

Methods:

Line S28: The authors refer to novel statistical methods without describing them and only citing unpublished work. Please provide more details and context of the novel modifications to the statistical approach because readers have no access to the reference.

I would find it informative to see some example plots in the supplementary information to help visualize the statistical techniques as the authors find appropriate (e.g. the b-splines and FPC scores for a sample time series).

L S62-63: Light extinction influences surface temperature but varies widely between lakes. Can you provide justification for using the extinction value of 1 m^{-1} in the Flake simulations?

Please provide details of the flake setup, eg was the model calibrated in any way or used ‘out of the box’? Did you also parameterize the fetch?

How did you account for ice cover in estimates of lake surface temperature from satellite measurements?

The legend of supplementary figure 1 could be more detailed – state that the data come from satellite measurements rather than the model, state what the median refers to (e.g. annual mean surface temperature).

A more informative explanation would also be helpful in the legend of supplementary fig. 3, since initially it reads as if lake elevation influences the latitude of a lake.

RShiny tool: in the online version there appear to be a large number of lakes in the Sahara (not in the manuscript) – consider removing them.

Caveats: I feel well qualified to assess the modelling, the analysis of surface temperature, and relationships with lake ecology. I have some limited experience in working with satellite-derived skin temperature data. I have experience in multivariate analyses and splines but not specifically in K-means clustering or QDA. Note that some of the statistical methods were not described in enough detail to allow an assessment (see comment above).

With best regards,

Tom Shatwell

Reply to referees NCOMMS-19-27289-T

Reviewers' comments:

Reviewer #1 (Remarks to the Author):

There is currently no global lake classification scheme for temperature, perhaps the most important variable in ecosystem function for lakes. This knowledge gap is probably because such an exercise is more complex than simply relating air temperature to water temperature. Here the authors have included not only average water surface temperature but also the seasonal dynamic, which is critical for understanding ecosystem function. Temperature and the seasonal patterns of turnover and stratification have broad and profound effects on biogeochemistry, gas flux, trophic relationships, organismal metabolism, and everything else that readily comes to mind in limnology. So the subject matter here is important, and the result is exciting. Further it's very useful to see that the authors have 1) used the classification scheme to consider climate scenarios, and also 2) demonstrated that small lakes can be correctly placed in ecoregions using extrapolation from the satellite data. The restriction of satellite data to large lakes has been a major criticism of its use. Thus the authors have filled a knowledge gap that will be of wide interest for the aquatic science/management community. I would hope that it would also be noted by other environmental scientists studying climate change who need to incorporate water bodies in their analyses, e.g. for gas flux and atmospheric dynamics.

We thank the reviewer for these positive comments and also for the helpful suggestions that we have used to improve the readability and impact of the manuscript.

I have 3 major suggestions, and various suggestions by line number.

1) Given that the research community at large could indeed use this work to feed global or regional analyses well outside of limnology – e.g. for atmospheric dynamics, or gas flux – researchers would need to trust wholly that the analyses are robust, such that they could view it as a product ready to plug and play, without having detailed limnological expertise themselves. In the Supplementary Methods (Line 28) it looks like the novel methods for accounting for periods of ice cover were not included for the reviewers, and this is a concern. If Gong et al. (in prep) methods are rejected by reviewers, or subsequently revised, what are the implications for this lake classification scheme? Ice covered lakes are indeed those anticipated to experience greatest change (Northern Frigid lakes on main text line 125-126). I have no way of evaluating the seriousness of this concern with the material at hand.

Thank you for this comment. We have now included additional information in the manuscript to address this issue. The added sentences are: 'Smoothing functions were fitted to each lake-mean time series to capture the major temporal patterns in the data using a saturated b-spline based statistical modelling approach that places a knot at every time point. When ice was detected, either by reflectance imagery or because of sub-freezing surface temperatures, the water below the ice was set to 0°C. To accommodate the discontinuities in the smooth time series that result from ice cover, the coefficients of the basis functions centred on the knots that

correspond to times with ice cover were set to zero. This effectively drops out the basis function whenever ice is present. This approach ensures that the number of basis functions used to estimate the underlying curve is the same for each time series which is necessary for the functional clustering of the curves.'

2) The authors could usefully place their findings in the context of those of O'Reilly et al. 2015 which produced groupings based on temperature trends, or Sharma et al's 2019 evaluations of ice cover (fig. 1) and projections of ice loss (fig. 2). Currently those papers are referenced but no explicit comparisons of results are made that would synthesize this paper's findings with those of previous work. Were the groupings or other spatial patterns similar? Did groupings of this paper map onto long term trends shown in O'Reilly et al (e.g. their figure 3)? In particular, O'Reilly et al's finding that warming was not predictable by region, lake depth, or air temperature trends, but related additionally to cloud cover and radiation trends, together might suggest that the projections of ecoregion change in this paper will be in error. In a practical sense, some of the highest and lowest warming rates in that paper occurred in lakes that were of similar elevations and air temperatures but with divergent cloud cover or solar radiation temporal trends. Data from both papers are all public so there really is not much constraint on how far the authors could take a synthesis.

We thank the reviewer for this suggestion. We have aligned the data for the 235 lakes in the Supplementary Table 4 in O'Reilly et al. (2015) with the thermal regions defined here. Although the eight regression tree leaves in O'Reilly are defined by a range of climatic, geographic and morphometric factors, there was only a weak correspondence between these clusters based on change in summer temperature and the thermal regions: a particular 'leaf' was distributed across three to eight thermal regions. This is concordant with the finding of O'Reilly et al. that there was not a strong regional pattern in their regression tree leaves. We have briefly commented on this in the text and provided a supplementary figure presenting the analysis.

The suggestion to synthesise data from a range of studies and measurement techniques is an excellent one and we hope that the work presented here will be a valuable framework to make this possible in the future (see final comment from this reviewer). However, given the very different spatial, seasonal and long-term differences between the data analysed here and in O'Reilly et al (2015) and Sharma et al (2019) a considerable amount of work would be required to align these disparate data sets and feel that this is better done in a separate paper.

With regards to the points raised on the future projections being in error, we emphasize that our model projections used as inputs to FLake, a wide range of meteorological information including those found to be important in O'Reilly et al.: cloud cover and radiation, as specifically mentioned by the reviewer, in both the historic and future periods.

3) Several of my comments are meant to improve the reader's understanding of the motivation for the paper. Right now the text does not establish the motivation very strongly - this weakness undermines the important messages in this paper. The summary is the primary section needing revision in this respect.

Thank you for bringing this to our attention. We have now improved the summary in-line with the reviewers suggestion.

Line 23- the reader doesn't know what a lake thermal ecoregion is – limnologists don't currently have them, and may not have thought much about needing them

In line with other reviewers' comments, we have changed 'thermal ecoregion' to 'thermal region' which we believe is intelligible to a general reader in the Abstract without further qualification.

Line 24 – the reader doesn't know what an ecoregion framework is yet, so using it as a baseline is pretty confusing

See above and we have had to shorten the abstract to fit the journal requirements and now do not use this word.

Line 32 – RCPs – this is not familiar terminology to many aquatic scientists – something like “recent scenarios of emissions” might get to the point more quickly

We understand and appreciate the reviewers concern, and have altered the text as a result. We have altered the relevant sentence to 'Using a lake model forced by 21st century climate projections we project that 12, 27 and 66% of lakes will change to a lower latitude thermal region by 2080-2099 for low, medium and high greenhouse gas concentration trajectories (Representative Concentration Pathways 2.6, 6.0 and 8.5 respectively).'

Line 97-99 – the result that water temperature doesn't track air temperature neatly is very similar to a main finding of O'Reilly et al. 2015 (Figure 2), one of several places in this manuscript where this work could be more soundly placed in the context of previous global analyses.

Thank you for this suggestion. We agree that it is important to place our results in the context of other global lake studies wherever possible. We have now made reference to the O'Reilly et al paper in the updated version of our manuscript.

Line 115 – to say that O'Reilly et al. found regional variations is somewhat misleading – the trends didn't typically clump by region but by a variety of regional climate and geomorphic factors – many of the groupings were dispersed throughout the world rather than by region (e.g. Fig 3 <https://agupubs.onlinelibrary.wiley.com/doi/full/10.1002/2015GL066235>).

We had not meant to mislead so thank you for mentioning this. We now have written the following: 'Analysis of historic data has shown that summer lake surface temperature is increasing globally but varies spatially depending on lake characteristics and regional changes in climate.'

Line 122-125 – this sentence doesn't quite make sense – the lakes don't move. Something more like... “The ecoregion status of a lake will shift toward those that are currently represented at lower latitudes...”? It's important to get this statement right since readers need to think about high latitude lakes becoming more like low latitude lakes over time, in order to grasp the concept and consider the implications.

We understand the motivation for this and we accept that we should be more precise. We have changed the sentence to the following: ‘There will be major changes in the global distribution of lake thermal regions by 2080-2099 (Fig. 4). Higher latitude lakes will tend to become like lower latitude lakes. The frequency of these shifts is projected to increase with the severity of climate change, being 12%, 27% and 66% for RCP 2.6, 6.0 and 8.5 respectively. For RCP 8.5, the number of Northern Frigid lakes will decline by 79%, while the number of Northern Temperate and Northern Warm lakes will increase by 166% and 228% respectively. The status of a total of 13,715 lakes (1%) will shift by more than one thermal region and an additional 48 Northern Cool lakes will shift status by three thermal regions to become Northern Hot lakes..’

Line 127 – use of the “lakes... move” again – I’d suggest rethinking this language. The reader is in a spatial state of mind for this paper, so “move” conjures up an image of the lake moving on a map. **See above.**

Line 130-32 – I have to think about it pretty hard to understand this statement that “seasonal surface temperature dynamics among lakes are essentially a continuum”. Is this a really important concept to convey? If so, I’d suggest rephrasing.

We agree and have simply deleted this preliminary phrase. The sentence now reads ‘We show here that robust, statistically-derived lake thermal regions can be produced.’

Line 136-138 – this is an important starting point but for many of these issues we will need to know more about lake morphometry, connectivity with other surface and ground waters, biodiversity, and probably other components that will affect refugia and biogeochemistry. A big unknown when working with satellite data is the focus on surface temperature and an inability to infer important characteristics of stratification and deep water dynamics, let alone biodiversity that is not observable at the surface. While I don’t think the authors need to caveat it to death, some acknowledgements of these important limitations should be included. Personally I am always concerned that the erroneous impression that satellite-based modeling can capture all of the important variables now when they actually capture relatively few aspects will lead to undesirable outcomes – e.g., that in situ monitoring programs will be terminated by decision makers, or well-meaning graduate students will restrict their analyses to satellite data and associated modeling rather than understanding limnological data in 3 dimensions. (Maybe someday, but right now we are not there.)

We completely agree with this and see remote sensing data as complementary to other types of data. We have added the following sentence. ‘Synoptic data derived from remote sensing are powerful approaches to characterise large scale patterns in surface conditions of lakes. However, a full assessment of climate change effects on lake structure, function and conservation value will require additional data derived from traditional and high-frequency measurements of conditions over the full depth of a lake. This will be especially the case for specific lakes or lake regions where an additional range of attributes including the connectivity between the lake and its catchment, lake bathymetry, and biodiversity especially for key species that are ecological engineers or of high conservation or commercial value will be required’

Figure 2 – it looks like grey not black indicates missing data

The Figure legend has been altered to note that grey areas are where lakes are absent from the database.

Figure 3 – without a legend for all the detail on these axes, panels b and c aren't useful, and panel a is marginally more interpretable. A 1:1 line would be useful in panel D where one could think a little more about ecoregions where small lakes might not map as well.

To help interpretation of the axes in the legend we have given references for panels a and b and referred to a new table in supplementary information that 'translates' the labels fully. For panel c, the different groups have no particular meaning, but the groups are described in the Methods. We have tried adding a 1:1 line on panel d as suggested but did not feel this helped the interpretation so have not implemented this.

Figure 4 – in the caption, it would be useful to note that black on the middle column is 1:1. What does black mean on the far right column? Presumably grey on the far left means missing data?

Thank you. We have now clarified this in the legend.

Supplementary Fig. 2 – was coherence quantified? If not, this figure does not literally show coherence, but it shows temporal patterns of lakes within each grouping (and the reader may visually infer some coherence).

Coherence can have different meanings but here we mean synchronous patterns in time series- we use this phrase now instead of coherence.

Supplementary Fig. 3 and Line 62-66 – Lake elevation can't affect latitude of a lake – the caption needs to be revised somehow. I've been staring at the figure for some time now and the message remains unclear. I find the reference to the Figure in the main manuscript line 62-64 equally perplexing.

We understand why this seems confusing and have rewritten the legend to try to make it clearer. One reason for plotting the data this way around is because the y-axis is more intuitive as it runs from north to south. The new legend reads: 'Effect of lake elevation on the latitude at which lakes in each of the nine thermal regions are found. For a given thermal region, lakes at high elevation can occur closer to the equator than lakes at low elevation.' We have altered the text to try to clarify and focus more on the thermal region rather than an individual lake: 'Lake elevation had an effect on the distribution of the thermal regions: for example, the two most northerly lake thermal regions, Northern Frigid and Northern Cool, were also found at high elevation in the Tibetan Plateau at a relatively low latitude. Accordingly, apart from Northern Warm, a given thermal region occurred nearer the equator at high elevation (Supplementary Fig. 3).'

Altogether this manuscript represents a lot of work and I will be excited to see how the research community can build upon it!

Yes we hope so too!

Reviewer #2 (Remarks to the Author):

In general, lakes are considered to reflect their watershed, and have been loosely characterized based on their terrestrial biomes (e.g. alpine lakes), lake-specific characteristics (e.g. shallow lakes), or their water quality (e.g. eutrophic lakes). The paper proposes a quantitative approach that can characterize lakes into ecoregions based on their thermal properties (mean and variation). In itself, that is somewhat novel – ecoregions that are defined purely based on mathematics. In fact, it is quite astounding how distinct these patterns appear to be (Supplementary Fig 2). The broader community would find this interesting in a theoretical context at the very least.

The level of detail provided appears to be adequate for reproducibility and many of the analytical tools are accessible. I appreciated the efforts to make these results accessible and applicable via the online app. However, I could not get the online app to load, and this is obviously an important component of broader implications and outreach associated with this work.

Thank you for these comments. However, because of major issues with potential maintenance of the app as software and packages are updated we have decided to rely solely on the code deposited in GitHub as this will provide greater future longevity and stability.

Ultimately, I was comfortable with the analytical approach. Initially, I was concerned that the long term mean temperature was not a good approach given that other studies have shown that lakes have been warming over time, at least seasonally. However, the approach seems robust. Many of my questions were adequately addressed in the Supplementary Information. Although the ice-detection method is still in the process of peer review, I recognize that with 2 temperature measurements each month, the temporal resolution is at the scale where error in ice detection is unlikely to have a substantial impact on the long-term mean.

We are happy that the reviewer is comfortable with the analytical approach we followed in our study. We have provided updated information in the manuscript with regards to the ice detection and its influence on the clustering algorithm. However, as the reviewer states this should have minimal influence on the results.

I was a bit concerned about accuracy of the projections. Most lakes are warmer than air (as can be seen in SI Fig 5) and other studies have found that many lakes are warming faster than air temperatures. Solar radiation is a major contributor to lake temperature, and global brightening and dimming is a factor contributing to lake warming trends. I wondered to what extent solar radiation has been factored in? If the projections are based solely on air temperature projections, this needs to be clear in the main manuscript and there should be some discussion about other contributing factors. I do not think that air temperatures are really that good of a predictor of lake temperatures.

We very much agree with the reviewer. To simulate lake temperature responses to climate change, we use a lake model forced by projections of air temperature at 2 m, wind speed at 10 m, surface solar and thermal radiation, and specific humidity. Thus, changes in solar radiation,

which as the reviewer rightly states are very important, are included in the results. We have now made this clearer in the text.

Ultimately, while I was comfortable with the analytical approach that was used here, it is fundamentally a statistical approach to defining ecoregions. I was left wondering if it really held up with respect to the actual ecology of these systems. For example, in terrestrial biomes, temperate grasslands and deciduous forests have similar mean temperatures but different amounts of precipitation, and the ecological structures and communities are fundamentally different. In these lakes, are there real differences between the southern and northern warm lakes when the statistical differences are only with respect to seasonal temperature variation? On one hand, I was looking for more of this type of exploration into ecosystem structure and function; on the other hand, this paper provides a framework that would allow hypotheses about differences and similarities to be addressed in future work (and perhaps these authors are already considering this as a next step). At the same time, these lake ecoregions are not as useful as being proposed here if they don't clearly translate into meaningful ecological differences. At the very least, I am not sure if the term 'ecoregion' is appropriate when only one variable (temperature) is explored.

Thank you for these comments and suggestions. Because another reviewer questioned the use of the term 'lake ecoregion' we have reverted to 'lake thermal region'. We do see this work as part of the process of providing a more synoptic view of lake structure and function and plan to include these thermal regions with other types of remotely sensed data as well as global biodiversity datasets to achieve this.

Supplementary Information

L62-63. Why was the light attenuation coefficient set to 1 m^{-1} ? How much difference does it make if a different value is used? (e.g. Just because it is widely done for some types of modeling doesn't necessarily mean it's also the best approach here.)

For the modelling we required a single value for light attenuation and so selected a moderate one and applied it to all the model runs. The good fit between model output and satellite observations suggested it was an appropriate choice, and the independent test of 71 smaller lakes indicated that the regions determined from the modelling were reasonable fits, giving further confidence in the approach. Previous work (Persson & Jones, 2008, *Freshwater Biology* 53, 2345-2355) suggested that the impacts of light attenuation on seasonally averaged temperature is relatively small, except for very clear, relatively small lakes. We have added a comment about this in the text and cited Persson & Jones (2008) and mentioned that transparency has an effect on mixing regime rather than surface temperature (Shatwell et al. 2019 op. cit.).

L 76. What data were extracted from ISIMIP for the projections – all variables or just temperature? **The data from ISIMIP needed to drive the lake model were: air temperature at 2 m, wind speed at 10 m, surface solar and thermal radiation, and specific humidity. We have added this detail to the methods.**

Reviewer #3 (Remarks to the Author):

This article presents a comprehensive and novel analysis of the global distribution of lake surface temperature characteristics. It's broader aim is to provide a framework to help interpret knowledge of lake ecology and effects of global change. It further claims that future climate warming will shift a certain proportion of lakes from a higher latitude to lower latitude ecoregion, whereby the northernmost lakes will be particularly affected. I think the conclusions are well supported.

The study is based on a representative global set of satellite-derived lake surface/skin temperature data and is combined with global modelling of lake surface temperature using an established physical model forced by state of the art meteorological data and climate projections. The authors also provide an online interactive tool to classify lakes into "ecoregions" according to the analysis in the manuscript. The code and data documenting the analyses are available on github, which enhances future use of the data and the classification scheme within the scientific community. Altogether the online material presents a useful package to help others build on the authors' work.

I think this is a good article and in my opinion it represents a valuable and potentially high-impact scientific contribution. The "ecoregions" provide a useful framework for interpreting impacts of climate warming. The paper is concise, well written and well structured. In some places it is too concise and I have described these below. The combined modelling and observed data analysis is a robust approach. The authors have made an effort to be complete by including a separate analysis of lakes too small to be included in the satellite data analysis. I found the consideration of the effect of lake elevation on the thermal classification necessary and informative. The "ecoregions" are highly correlated with latitude but the authors provided an analysis of the added information of their scheme compared to other classifications or simply latitude. The statistical analyses appear to be thorough and appropriate, though some information is missing. The most important references have been reviewed to my knowledge. Almost all of my comments are minor and related to clarity or a wish for more details. Most of these things can be dealt with in the supplementary material.

There is only one potentially significant issue, which is possibly just a misunderstanding. In the global modelling, the authors state in the methods that they used ERA-Interim reanalyses to force the lake model, which is an established and acceptable method to run lake simulations. However I see a potential issue with the comparison of climate warming scenarios (RCP 2.6, 6.0 and 8.5) with historical data derived from ERA-Interim reanalyses to draw inferences about the effect of warming. This would introduce bias into the results, and any shifts of lakes from one ecoregion to another may be due in part to this bias rather than due to warming alone, which would question some of the results. The valid approach is to compare the RCP warming projections with the historical scenario generated from the same global climate model. It is possible that this is what the authors in fact did because supplementary figures 6, 7, 8, and 9 show a "historical" scenario. In any case this is not clear to me and I ask that the authors clarify whether the "historical" data are derived from ERA-

Interim or from the respective climate model. If the historical scenario is based on ERA-interim, then additional work is required to deal with the bias.

We thank the reviewer for their positive comments and helpful suggestions. The final point and concern was about possible bias introduced by having two historical outputs, one from ERA-Interim and the other produced with the climate change scenarios. In fact we had done what the Reviewer has suggested was the correct approach and generated the historical scenarios from the same climate model used for the future scenarios: so the comparison is a fair one. We have emphasised this point in the Methods section, the text dealing with the results and in the legend to Fig. 4.

Other comments

I am questioning the use of the term (thermal) “ecoregion”, which implies that lakes are classified according to their ecology whereas in fact they are classified only according to surface temperature. Surface temperature alone is insufficient to classify lake ecology. The authors gave a nice summary of how the thermal classification can contribute to classification of lake biomes (L147-151) but perhaps a different term is more appropriate. If you feel differently, it would be helpful to establish a broad link between lake ecology type and the “ecoregions” according to surface temperature based on literature, for instance. This is a semantic issue and I am not questioning the approach in general.

This was questioned by another reviewer. On reflection we have decided to use the term ‘thermal region’ rather than thermal ‘ecoregion’. We have added a small amount of text suggesting that in the future this framework might be expanded to produce lake ecoregions or even biomes.

Would it be worthwhile to include a southern frigid region to include Antarctic lakes for the sake of completeness?

The geography of the southern hemisphere means that no Southern Cool or Southern Frigid lakes were identified. We understand the aspiration behind this question, however, since we used a data-driven approach it is not really possible to invent these two new categories.

I am confused about how lakes were classified according to location (e.g. L 113, Supplementary Table 2). It is probably quite simple but not obvious to me from the manuscript. Please provide details about how the criteria were derived to classify lakes according to geographic location.

Lakes were classified from their location based on the modelling outputs at a 2 degree grid. Thus there is a lake thermal region for each 2-degree grid- shown in Fig. 2c for example.

Could you comment on how you think the classification scheme works for brackish or saline lakes in arid or semi-arid regions? For instance many of the natural lakes shown in Australia are salty and/or even ephemeral. How did the lake model perform against satellite data in these regions?

We don’t have access to a global dataset of lake salinity but have now added the following sentence in the manuscript. ‘The 732 lakes with satellite-derived surface temperature data include lakes which are brackish or saline as well as fresh. Except that overlying meteorology will influence both the likelihood of salinity and the surface temperature cycle, we do not expect

impacts of salinity to supersede meteorological effects sufficiently to distort greatly the classification. In mainland Australia, for example, clustering analysis showed that all the lakes in the dataset fitted the Southern Warm class (Fig. 1e) despite having a large range of salinities.'

Do you know why southern hemisphere lakes show less variability at the same mean temperature than northern hemisphere lakes?

Yes. It results from the difference in the fraction of land mass distribution between the two hemispheres, which makes the SH landmasses strongly moderated by maritime influence on surface air temperature, because of the ocean's large thermal inertia compared to land (on seasonal time scales). Annual surface temperature range is substantially less on average across all SH land masses than corresponding latitudes in NH. Despite solar insolation ranges that are fairly hemispherically symmetric, the smaller air temperature range moderates the lake temperature range via lake-air heat fluxes driven by lake-air temperature difference. See Layden et al. (2015) op. cit. We have commented on this briefly in the manuscript.

L80: "was" not "were".

Corrected

L81: "all" repeated.

Corrected

L113: This sentence requires some more explanation. What is meant by the sub-dominant group?

We have rewritten the sentence: 'Eighty percent of the lakes were correctly matched to the same thermal region based on geographic location and this increased to 97% if the second closest match was included (Fig. 3d).'

L123: "projected" is more appropriate than "forecast"

We have changed the word.

Methods:

Line S28: The authors refer to novel statistical methods without describing them and only citing unpublished work. Please provide more details and context of the novel modifications to the statistical approach because readers have no access to the reference.

We have now done this. The added sentences are: 'Smoothing functions were fitted to each lake-mean time series to capture the major temporal patterns in the data using a saturated b-spline based statistical modelling approach that places a knot at every time point⁵. When ice was detected, either by reflectance imagery or because of sub-freezing surface temperatures, the water below the ice was set to zero °C. To accommodate the discontinuities in the smooth time series that result from ice cover, the coefficients of the basis functions centred on the knots that correspond to times with ice cover were set to zero. This effectively drops out the basis function whenever ice is present. This approach ensures that the number of basis functions used to estimate the underlying curve is the same for each time series which is necessary for the functional clustering of the curves.'

I would find it informative to see some example plots in the supplementary information to help visualize the statistical techniques as the authors find appropriate (e.g. the b-splines and FPC scores for a sample time series).

Thank you for these suggestions. We have added an example of the b-splines for an example time series (Supplementary Fig. 11) and provided a plot of the FPC scores (Supplementary Fig. 12).

L S62-63: Light extinction influences surface temperature but varies widely between lakes. Can you provide justification for using the extinction value of 1 m^{-1} in the FLake simulations?

For the modelling, we required a single value for light attenuation and so selected a moderate one and applied it to all the model runs. The good fit between model output and satellite observations suggested it was an appropriate choice, and the independent test of 71 smaller lakes indicated that the regions determined from the modelling were reasonable fits, giving further confidence in the approach. Previous work (Persson & Jones, 2008, Freshwater Biology 53, 2345-2355) suggested that the impacts of light attenuation on seasonally averaged temperature is relatively small, except for very clear, relatively small lakes. We have added a comment about this in the text and cited Persson & Jones (2008) and mentioned Shatwell et al. (2019) for an effect of transparency on mixing regime.

Please provide details of the flake setup, eg was the model calibrated in any way or used 'out of the box'? Did you also parameterize the fetch?

The FLake model was used 'out of the box' i.e., we did not calibrate the model parameters to improve the model fit. This is a major strength of the modelled temperatures (i.e., that the modelled temperatures simulated accurately the seasonal clusters without calibration). Fetch was estimated based on the surface area of the lakes (i.e., values extracted from the Hydrolakes database) - square root of surface area. We have explained this in the manuscript.

How did you account for ice cover in estimates of lake surface temperature from satellite measurements?

'When ice is detected (either by reflectance imagery or because of sub-freezing ice surface temperature) the water below the ice is set to 0°C .' We have added this sentence to the methods.

The legend of supplementary figure 1 could be more detailed – state that the data come from satellite measurements rather than the model, state what the median refers to (e.g. annual mean surface temperature).

Thank you for these suggestions- we have implemented them.

A more informative explanation would also be helpful in the legend of supplementary fig. 3, since initially it reads as if lake elevation influences the latitude of a lake.

We understand why this seems confusing and have rewritten the legend to try to make it clearer. One reason for plotting the data this way around is because the y-axis is more intuitive

as it runs from north to south. The new legend reads: 'Effect of lake elevation on the latitude at which lakes in each of the nine thermal regions are found. For a given thermal region, lakes at high elevation can occur closer to the equator than lakes at or below sea-level.'

RShiny tool: in the online version there appear to be a large number of lakes in the Sahara (not in the manuscript) – consider removing them.

We are not sure how this occurred but it has now been corrected in the GitHub files. However, we have reluctantly decided not to include the app because of major concerns about maintaining it in the future as systems and software are updated.

Caveats: I feel well qualified to assess the modelling, the analysis of surface temperature, and relationships with lake ecology. I have some limited experience in working with satellite-derived skin temperature data. I have experience in multivariate analyses and splines but not specifically in K-means clustering or QDA. Note that some of the statistical methods were not described in enough detail to allow an assessment (see comment above).

Further information is now provided on the statistical methods applied in this project.

With best regards,
Tom Shatwell

REVIEWERS' COMMENTS:

Reviewer #1 (Remarks to the Author):

The authors have strengthened the paper in response to the reviews, and I continue to see this work as an important contribution. The revision satisfies my concerns. The ms is now placed more firmly in the context of previous research. The unpublished Gong et al methods are helpfully described. The wording regarding the “movement” of lakes to different categories is improved. Caveats regarding remote sensing are included without becoming burdensome to the text. Figure legends are improved and corrected.

The abstract/summary still struck me as weak and now I think it’s just the first couple sentences. These first sentences are so important for engaging the reader, and particularly for reaching non-limnologists who I really want to be aware of this useful manuscript.

I don’t know how it fits the journal guidelines for this summary but I am looking for a beginning that is more like this:

“Water temperature dynamics are critical for key ecological processes in lakes. Thus understanding and predicting lake states and processes would be substantially aided by a thermal classification system, yet such a scheme is lacking.”

With this approach the motivation/importance is up front, understandable to non-limnologists, and the authors could eliminate or shorten the last sentence.

Line 279 has a typo

Line 289 – suggest “ a grouping of six clusters was...” , rather than “six clusters was”

Reviewer #2 (Remarks to the Author):

The authors have done a good job of responding to my concerns. I am more comfortable with the idea of thermal regions than with ecoregions. This both simplifies the message and simultaneously

raises new questions for future research. I think this work could spur a wide range of studies using more ecological indicators. My few remaining comments are listed below.

I am not clear on how my earlier comment on air temperature/solar radiation was incorporated into the manuscript. The authors' response seems to indicate that changes in solar radiation were included in the results, but 'solar radiation' is not specifically mentioned anywhere in the main manuscript. Would it be possible to list the key variables incorporated into the model in the main text (maybe into the sentence on line 76 "The module was driven by forcing data. . ."). This would achieve several purposes: 1) alleviate the need for a skeptical reader to go the Methods just to see what those variables were, and 2) illustrate all those driving factors for readers who might just be learning about what influences lake temperatures.

I was a bit confused by line 114 about the color scheme being in Supplementary Information . It might be helpful to leave the 'RGB' in that sentence.

I think it is a bit of a stretch to assign O'Reilly et al. 2015 as responsible for saying that "temporally changes in air and lake temperature do not track one another" (line 131). This was not the purpose of their study nor specifically what they said. Maybe use a different reference or rephrase this statement.

Line 146. Do you know how many of those lakes are not fresh? If so, it might be helpful to include the % of lakes to underscore that this is a minor issue (presumably).

Line 178. I think the percents here are referring to the proportion of lakes that will change, right? The sentence was not exactly clear.

I think you need to cite Ref #1 again after the sentence starting on line 213 "Instead . . ."?

What is the status of Gong et al. in prep? It would be really nice to include that reference as it presumably more fully describes the process of accounting for ice cover and it would be good to better make this connection to the more substantive work in the forthcoming Gong et al. paper.

Reviewer #3 (Remarks to the Author):

The authors have made a concerted effort to revise the paper, and sufficiently addressed all points I raised in the first review. The revised paper reads well. Except for the minor points below, I have no further comments.

Consider rewording the sentence on L110-112 to use spatial and temporal as adjectives rather than as adverbs (spatially...) which sounds a bit wordy.

Please check the legends of Supplementary Figs 10 and 11 for minor typos.

Sincerely,

Tom Shatwell

REVIEWERS' COMMENTS:

Reviewer #1 (Remarks to the Author):

The authors have strengthened the paper in response to the reviews, and I continue to see this work as an important contribution. The revision satisfies my concerns. The ms is now placed more firmly in the context of previous research. The unpublished Gong et al methods are helpfully described. The wording regarding the “movement” of lakes to different categories is improved. Caveats regarding remote sensing are included without becoming burdensome to the text. Figure legends are improved and corrected.

We thank the Reviewer for these comments.

The abstract/summary still struck me as weak and now I think it’s just the first couple sentences. These first sentences are so important for engaging the reader, and particularly for reaching non-limnologists who I really want to be aware of this useful manuscript.

I don’t know how it fits the journal guidelines for this summary but I am looking for a beginning that is more like this:

“Water temperature dynamics are critical for key ecological processes in lakes. Thus understanding and predicting lake states and processes would be substantially aided by a thermal classification system, yet such a scheme is lacking.”

We thank the reviewer for this suggestion and have taken and modified the idea. The first two sentences now read ‘Water temperature is critical for the ecology of lakes. However, the ability to predict its spatial and seasonal variation is constrained by the lack of a thermal classification system.’

With this approach the motivation/importance is up front, understandable to non-limnologists, and the authors could eliminate or shorten the last sentence.

We have shortened the last sentence.

Line 279 has a typo

Corrected.

Line 289 – suggest “ a grouping of six clusters was...”, rather than “six clusters was”

Changed.

Reviewer #2 (Remarks to the Author):

The authors have done a good job of responding to my concerns. I am more comfortable with the idea of thermal regions than with ecoregions. This both simplifies the message and simultaneously raises new questions for future research. I think this work could spur a wide range of studies using more ecological indicators. My few remaining comments are listed below.

I am not clear on how my earlier comment on air temperature/solar radiation was incorporated into the manuscript. The authors' response seems to indicate that changes in solar radiation were included in the results, but 'solar radiation' is not specifically mentioned anywhere in the main manuscript. Would it be possible to list the key variables incorporated into the model in the main text (maybe into the sentence on line 76 "The module was driven by forcing data. . ."). This would achieve several purposes: 1) alleviate the need for a skeptical reader to go the Methods just to see what those variables were, and 2) illustrate all those driving factors for readers who might just be learning about what influences lake temperatures.

We thank the Reviewer for these comments. Actually, solar radiation was specifically mentioned on L246 in the Methods section. However we take the point and have listed the driving variables at the location suggested.

I was a bit confused by line 114 about the color scheme being in Supplementary Information . It might be helpful to leave the 'RGB' in that sentence.

We have altered the sentence and mentioned RGB specifically as requested.

I think it is a bit of a stretch to assign O'Reilly et al. 2015 as responsible for saying that "temporally changes in air and lake temperature do not track one another" (line 131). This was not the purpose of their study nor specifically what they said. Maybe use a different reference or rephrase this statement.

O'Reilly et al 2015 said '*However, for individual lakes, air and lake temperature trends often diverged (Figure 2), emphasizing the importance of understanding the various factors that control lake heat budgets rather than assuming lake temperatures will respond similarly to air temperatures.- se we think our sentence was accurate. We have altered the original sentence slightly which we believe is correct: 'The low spatial agreement between surface lake water temperature and air temperature, matches the observed lack of temporal rate of change in air and lake temperature at individual locations¹.*

Line 146. Do you know how many of those lakes are not fresh? If so, it might be helpful to include the % of lakes to underscore that this is a minor issue (presumably).

Unfortunately there does not appear to be a global database with this information. Most of our lakes will not be saline but we cannot quantify this precisely.

Line 178. I think the percents here are referring to the proportion of lakes that will change, right? The sentence was not exactly clear.

This line number does not correspond to the submitted pdf so we are not sure what precisely is being referred to. We have assumed it was line 152 on the pdf and have altered this accordingly. The sentence now read, 'The frequency of these shifts is projected to increase with the severity of climate change: 12%, 27% and 66% of all lakes will move to a lower latitude thermal region for RCP 2.6, 6.0 and 8.5 respectively.'

I think you need to cite Ref #1 again after the sentence starting on line 213 "Instead . . ."?

Done.

What is the status of Gong et al. in prep? It would be really nice to include that reference as it presumably more fully describes the process of accounting for ice cover and it would be good to better make this connection to the more substantive work in the forthcoming Gong et al. paper. Unfortunately this is not yet submitted.

Reviewer #3 (Remarks to the Author):

The authors have made a concerted effort to revise the paper, and sufficiently addressed all points I raised in the first review. The revised paper reads well. Except for the minor points below, I have no further comments.

We thank the Reviewer.

Consider rewording the sentence on L110-112 to use spatial and temporal as adjectives rather than as adverbs (spatially...) which sounds a bit wordy.

We have rewritten the sentence which now reads, 'The low spatial agreement between surface lake water temperature and air temperature, matches the observed lack of temporal rate of change in air and lake temperature at individual locations'.

Please check the legends of Supplementary Figs 10 and 11 for minor typos.

Corrected.

Sincerely,
Tom Shatwell